# Cross-domain correspondence intensity modulation based on Bayesian-decision for remote sensing image pansharpening

**Lei Wu** ⓘ *, **Xunyan Jiang, Zhijian Zhao, Zhaosheng Xu, Jinhua Liu**

Xinyu University, College of Mathematics and Computer, Xinyu, China

* jxxywulei@126.com

## Abstract

Pansharpening usually improves the resolution of low-resolution multispectral (LRMS) images with spatial information from corresponding high-resolution panchromatic (HRPAN) images to produce high-resolution MS (HRMS) images. Traditional pansharpening methods use various domain transformations to make the fused image suffer varying degrees of spatial or spectral distortion because the information in the LRMS and PAN images is heterogeneous and distributed in different domains. The motivation of our proposed work is to develop a balanced and robust pansharpening method named cross-domain correspondence intensity modulation, which is based on Bayesian decision-making for remote sensing image pansharpening. First, the intensity component of the MS image is obtained via the intensity hue saturation (IHS) transform. Second, a fusion rule based on the Bayesian probabilistic model is designed to fuse the intensity component and the corresponding PAN image to obtain an intermediate component. Third, a cross-domain correspondence intensity modulation algorithm is proposed to modulate the intensity information in the intermediate component to produce the desired intensity component. Finally, an inverse IHS transformation is performed to obtain the pansharpened MS image by replacing the original intensity component with the modulated intensity component. The results on different satellite datasets show that the proposed method can effectively enhance the spatial and spectral fidelity of the fused image.

## 1. Introduction

At present, due to technical limitations, many multisatellite earth observation constellations provide high-resolution PAN (HRPAN) images and low-resolution MS (LRMS) images with the same scene at the same time and cannot provide high-resolution MS (HRMS) images [1]. Many application areas, such as crop phenology, change detection, military reconnaissance, and visual interpretation, require HRMS images [2]. HRPAN and LRMS image synthesis is the best solution for overcoming this problem.

**Data availability statement:** All relevant data are within the paper and its Supporting information files.

**Funding:** This work is supported by National Natural Science Foundation of China (No. 62441112 and No. 72461031), by Jiangxi Provincial Natural Science Foundation (No. 20232BAB201026), and by Science and Technology Re-search Project of Jiangxi Provincial Department of Education (No. GJJ2402102).].

**Competing interests:** The authors have declared that no competing interests exist.

Pansharpening is an artificial information merging technology that is a research hotspot in HRMS image acquisition [3].

In the past 20 years, with the development of remote sensing technology and remote sensing image applications, different pansharpening methods have been developed. These methods can be divided into two main categories: traditional pansharpening methods and deep learning (DL)-based methods. Currently, the latter is the most popular method. Lu *et al*. [4] proposed a multiscale self-attention net-work (MSAN) to integrate local and long-range features of remote sensing images for pansharpening, in which revised Swin Transformer block is developed for the preservation of spectral-spatial information. Song *et al*. [5] developed an invertible attention-guided adaptive convolution and dual-domain Transformer (IACDT) network to correct locally misaligned features by integrating their long-range dependencies for pansharpening. Lu *et al*. [6] developed a lightweight pansharpening network, in which, a cross-scale interactive encoder was proposed to improve scale consistency for pansharpening. These methods exhibit excellent performance in remote sensing image pansharpening because of that convolutional neural networks (CNNs) have strong image processing and analysis capabilities. However, this class of methods learns the nonlinear relationship between the observed value and the pansharpened HRMS image, relying on large-scale datasets. The calculation is complicated, and the resource consumption is high [7].

To date, traditional pansharpening methods have dominated remote sensing fusion for a long time. Generally, methods can be divided into three main categories: component-substitution (CS)-based methods, methods based on multiresolution analysis (MRA), and model-based methods. Among them, the CS- and MRA-based methods are two types of conventional pansharpening methods. CS-based pan-sharpening methods break the MS image down into various components via a trans-formation [8]. Intensity−hue−saturation (IHS) is a widely used CS transformation that separates a red–green-blue (RGB) image into its three components, I, H, and S [9]. The I component possesses most of the spatial information from the MS image. The isolated spectral information is in the H and S components. PCA is another conven-tional CS technique that transforms MS images into various uncorrelated variables [10]. The information highly relevant to the MS image is contained in the first princi-pal component. In addition, the brovey transform (BT) method [11] is a type of CS method. Generally, the abovementioned CS methods need to explicitly compute the forwards and backwards transformations where the component containing the spatial information is replaced with the PAN image. As a result, the degree of distortion of the pansharpened image depends on the degree of relevance between the replaced component and the PAN image [12].

MRA methods are approaches based on multiscale decomposition for pansharp-ening. Pyramid transforms (e.g., Laplacian pyramids [13]) and discrete wavelet transforms (e.g., contourlet transforms and shearlet/wavelet transformations [14]) are the most widely used MRA methods. These methods commonly involve three steps: 1) MS images are decomposed into various subimages at several scale levels or different orientations; 2) the subimages at each level or orientation are fused; and 3)

an inverse transform is implemented to synthesize the fused image. Recently, spatial filtering (e.g., bilateral/guided filtering [15]) and sparse representation (e.g., dictionary learning and robust sparse representation [16]) have been favoured by scholars in the MRA-based pansharpening field. Lu et al. [17] proposed an intensity mixture and band-adaptive detail fusion method based on a filter estimation algorithm, in which, an intensity mixture model is designed to obtain a mixed-intensity image. Generally, these technologies adopt a very natural method to provide the required image characteristics. They are applied to extract high-frequency spatial information from source images at various scale levels [18]. Compared with the CS method, MRA methods can effectively preserve the spectral information in MS images, but the spatial information enhancement capability is worse than that of the CS method [19].

To improve the spatial and spectral quality of pansharpened images, model-based methods have been developed in recent decades. Generally, model-based methods first obtain the spatial and spectral priors from the source images on the basis of CS or MRA approaches. Fist, a model based on the priors is constructed. Second, a certain optimization algorithm is designed to solve the model. For example, Wu et al. [20] formulated a super-resolution model for low-rank tensor completion where the characterization of MS image was explored by introducing low-tubal-rank prior and the spatial information was captured via detail mapping based on local similarity. Wu et al. [21] developed a pansharpening model based on sparse reconstruction with semi-framelet-guided, in which, an algorithm based on proximal alternating minimization was designed to solve the proposed model. Wu et al. [22] explored learnable nonlinear mapping (LNM-SF) for spatial fidelity of remote sensing images, and a new variational model based on LNM-SF for apansharpening was developed. Recently, the most commonly used pansharpening model is the high-frequency detail injection (HFDI) model, where the spatial details extracted by CS or MRA technologies from the PAN image are injected into the MS image. Yang et al. [23] adopted guided filtering to extract high-frequency details that were highly correlated with the source MS image from the PAN image and injected the details into the LRMS image according to a proposed HFDI model with the help of an improved injection gain. Yang et al. [16] used sparse representation theory to reconstruct the spatial details of the MS image by constructing a dictionary to learn the spatial information of the PAN image and then injected the reconstructed details into the LRMS image to obtain the pansharpened image. Detail extraction and injection steps are essential in the HFDI model, the resulting images produced by the pansharpening methods based on the HFDI model are prone to spatial distortion. To improve the pansharpening results, Lu et al. [24] considered detail correction between PAN and MS images to calculate band-adaptive gradient to develop a unified pansharpening model for pansharpening. Wu et al. [25] proposed spectral intensity modulation based on a multiobjective decision for pansharpening. In [26], the theory of linear algebra based on a judgement matrix was applied to calculate the modulation coefficient such that state-of-the-art performance was achieved. Owing to the strong ability of spectral intensity modulation in image pansharpening, a pansharpening method based on cooperative representation was proposed [26]. The above methods show that the HFDI model can reduce image distortion effectively, but its effectiveness depends on the quality of the designed coefficients.

The fast multiband image pansharpening based on a probabilistic model (e.g., Bayesian/Markov probability model) determine the likelihoods of the observations that can be generalized to incorporate prior information for the pansharpening problem. For example, a hierarchical Bayesian model based on an appropriate prior distribution was developed to fuse multiple multiband remote sensing images in [27]. Yang et al. [28] proposed an efficient and high-quality pansharpening model based on the Markov maximum likelihood estimator. Wu et al. [29] exploited geometrical considerations and the posterior distribution of remote sensing images to propose a pansharpening model based on a Bayesian decision. The above methods show that no additional transformation is required in pansharpening methods based on a probabilistic model, which can effectively prevent the loss of image spatial information and maintain image spectral information.

In summary, there are differences in spatial and spectral resolutions between the PAN and LRMS images obtained by satellite-borne sensors. Spatial resolution refers to the amount of spatial detail that can be perceived in an image. The spectral resolution describes a sensor's capacity to gather data at a certain wavelength. Therefore, the PAN and MS images are heterogeneous and have different domains. The abovementioned pansharpening methods merge the

information between heterogeneous images in different domains, which causes spatial or spectral distortion of the pan-sharpened images. To address this problem, in this paper, cross-domain correspondence intensity modulation based on Bayesian decision-making for remote sensing image pansharpening is proposed. The proposed method uses pansharpening theory to synthesize the space-spectrum information of two images at corresponding positions in different domains while modulating the intensity information in the same domain to reduce image distortion. The main contributions of this work are threefold:

1) We construct a bayesian probabilistic model for image component fusion where the prior information on the basis of the I component obtained by an IHS transformation and the PAN image is calculated.

2) We design a fusion rule based on the Bayesian probabilistic model to obtain an intermediate component with complete spatial information of MS and PAN image domains, but missing the interdomain relationship.

3) We design a cross-domain correspondence intensity modulation algorithm to optimize the intermediate component to produce the desired I component to replace the original I component for the inverse IHS transformation to obtain the pansharpened image.

This paper is organized as follows. The relevant work is briefly described in Section II. The scheme of the proposed approach is presented in Section III. Section IV provides the experimental data settings and quality assessment indices. Experimental results and discussion are given in Section V. Finally, the conclusions are given in Section VI.

## 2. Proposed approach

In this work, we consider cross-domain fusion of information between heterogeneous images to obtain a desired I component. In the proposed method, an IHS transformation is used to obtain the original I component of the MS image and synthesize this final fusion image. The Bayesian probabilistic model is designed to create a fusion rule to obtain an intermediate component by choosing pixels from the original I component of the MS image and the PAN image. A cross-domain correspondence intensity modulation algorithm is designed to improve the intermediate component to produce the desired I component. The framework of the proposed method is illustrated in Fig 1.

In Fig 1, first, the LRMS image is upsampled (4×4) and interpolated to produce a UPMS image whose size is the same as that of the corresponding PAN image. Second, the $I$, $H$ and $S$ components of the UPMS image are obtained by an IHS transformation. Third, we construct a Bayesian probabilistic model based on the $I$ component and the histogram-matched

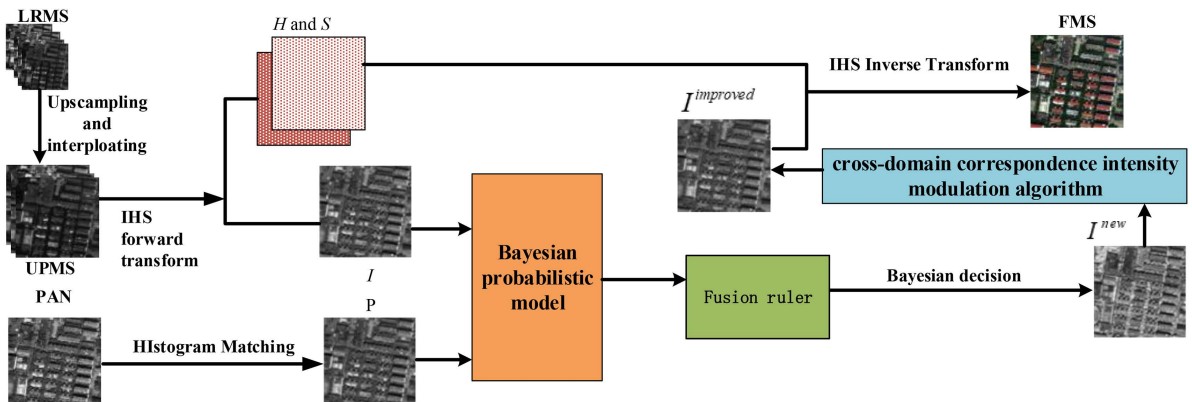

**Fig 1. Flowchart of the proposed approach. UPMS is the upsampled MS image.** IHS is IHS transformation, H and S is hue and saturation component, $I^{new}$ is the new $I$ component, $I^{improved}$ is the improved $I^{new}$, FMS is the fused image.

PAN image *P*. Furth, we believe that Bayesian decision-making can choose the desired pixels from the current image to obtain a new *I* component *I*$^{new}$, but it does not address the luminance relation between different domain pixels. Therefore, we design a cross-domain correspondence intensity modulation algorithm to improve *I*$^{new}$ to obtain the improved *I*$^{improved}$ component. Finally, the fused image is produced by replacing *I* with *I*$^{improved}$ in the IHS inverse transform. The detailed information for the proposed method is described in detail below.

## 2.1 Bayesian probabilistic model for image component fusion

The aim of this study is to find a desired I component to replace the original I component of the MS image to produce an HRMS image with spatial and spectral fidelity. Thus, we first obtain the original I component of the MS image. The simplest method is the IHS transformation, described as follows:

$$[I, H, S] = IHS(UPMS)$$

(1)

where *UPMS* is the UPMS image and where $IHS(\cdot)$ is the forwards IHS transformation.

Many studies have shown that traditional pansharpening method based on IHS transformation can effectively enhance MS images spatially, but it results in severe spectral distortion of the pansharpened image. To solve this problem, many scholars use different methods that fuse the PAN image and the I component of the MS image to improve the I component. The experimental results revealed that the resulting image generated via Bayesian decision-making can effectively enhance the image spatial information while reducing the degree of spectral distortion. In particular, the key is how to choose pixels from the PAN image and the *I* component of the MS image to construct *I*$^{new}$. In preliminary work [28], we proposed a fusion algorithm based on the Bayesian decision using the framework described in part II to merge the PAN image and the I component of the MS image to obtain a desired new I component. The flowchart of the Bayesian probabilistic model for image fusion is illustrated in Fig 2.

As shown in Fig 2, we designed a novel fusion rule based on Bayesian decision-making to integrate the PAN image and the *I* component of the MS image to obtain a new intensity *I*$^{new}$ component and replace *I* with *I*$^{new}$ to obtain the desired HRMS image. We calculated the probability that the pixel in the PAN image or the component of the MS image was selected on the basis of the Bayesian formula and selected a pixel with high probability to constitute the new *I*$^{new}$ component. Therefore, three categories of probabilities, namely, prior, conditional and posterior probabilities, must be calculated after an IHS transformation is applied to the original low-resolution MS image. As in [28], in this study, we assume

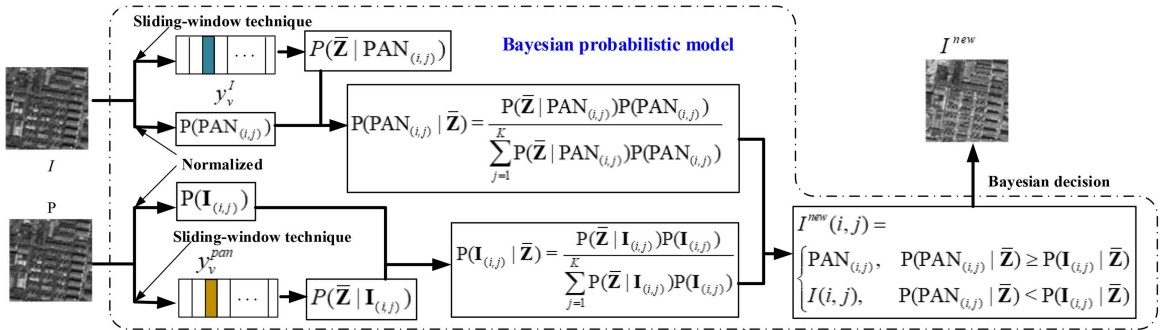

**Fig 2. Flowchart of Bayesian probabilistic model for image component fusion.** $\bar{\mathbf{Z}}$ is the class of the fused component, $P(PAN_{(i,j)})$ and $P(\mathbf{I}_{(i,j)})$ are the prior probability of pixel at coordinate $(i,j)$ in PAN image and the I component, respectively, $y_v^{pan}$ and $y_v^I$ are the $v$th image patches in PAN image and *I* component, respectively, $P(PAN_{(i,j)}|\bar{\mathbf{Z}})$ and $P(\mathbf{I}_{(i,j)}|\bar{\mathbf{Z}})$ are the posterior probability of pixel at coordinate $(i,j)$ in PAN image and the *I* component, respectively.

that each pixel is independent and that their prior probabilities rely on the principle of sensor imaging. Thus, the pixel values normalized to the PAN image and the *I* component are treated as the prior probabilities of the pixels, described as $P(PAN_{(i,j)})$ and $P(I_{(i,j)})$, respectively, where $PAN_{(i,j)}$ and $I_{(i,j)}$ are the pixels at coordinate $(i,j)$ in the PAN image and the *I* component, respectively. Furthermore, according to the theory that the change in intensity of a single pixel causes the spectral characteristics of its neighbouring pixel [18], we calculate the conditional probability such that a pixel adjacent to those pixels under the neighbouring pixels are the determination conditions. Let $y_v^{pan}$ and $y_v^I$ represent the image patches where adjacent pixels of $PAN_{(i,j)}$ and $I_{(i,j)}$ are located, respectively. We use a sliding-window technique to obtain the corresponding neighborhood pixels of $PAN_{(i,j)}$ and $I_{(i,j)}$ from the PAN image and I component. The number of neighboring pixels depends on the size of the sliding window. In this study, the sliding-window with overlapping areas of size $3 \times 3$ is used to divide the PAN image and the *I* component of the LRMS image with size of $m \times n$ into V patches: $\{y_v^{pan} \mid v = 0, 1, ..., V-1\}$ and $\{y_v^I \mid v = 0, 1, ..., V-1\}$ from top to bottom. The treatment of the sliding window technique (take the PAN image as an example) can be seen as Fig 3.

Let $\bar{Z}$ be the class of the fused component. Then, the following Eq. (2) and Eq. (3) in this work calculate conditional probabilities $P(\bar{Z} \mid PAN_{(i,j)})$ and $P(\bar{Z} \mid I_{(i,j)})$, which are the probabilities of a pixel in the PAN image or the *I* component of the MS image being selected to construct $I^{new}$ when the natural state of the PAN and LRMS observed image occurs.

$$P(\bar{Z} \mid PAN_{(i,j)}) = \prod_{v=1}^{V} P(\bar{Z} \mid y_v^{pan}) = P(\bar{Z} \mid y_{v,(1,1)}^{pan}) \times P(\bar{Z} \mid y_{v,(1,2)}^{pan}) \times ... \times P(\bar{Z} \mid y_{v,(r,s)}^{pan}) \tag{2}$$

$$P(\bar{Z} \mid I_{(i,j)}) = \prod_{v=1}^{V} P(\bar{Z} \mid y_v^I) = P(\bar{Z} \mid y_{v,(1,1)}^I) \times P(\bar{Z} \mid y_{v,(1,2)}^I) \times ... \times P(\bar{Z} \mid y_{v,(r,s)}^I) \tag{3}$$

where $P(\bar{Z} \mid y_{v,(r,s)}^{pan})$ and $P(\bar{Z} \mid y_{v,(r,s)}^I)$ are the prior probabilities at coordinate $(r, s)$ in a window of the PAN image and the *I* component, respectively. The corresponding computational process can be seen as Fig 4.

According to Bayes theory, the posterior probabilities of the pixels in the PAN image and the *I* component can be given by

$$P(PAN_{(i,j)} \mid \bar{Z}) = \frac{P(\bar{Z} \mid PAN_{(i,j)})P(PAN_{(i,j)})}{\sum_{j=1}^{K} P(\bar{Z} \mid PAN_{(i,j)})P(PAN_{(i,j)})} \tag{4}$$

**Fig 3. The treatment of the sliding window technique (take the PAN image as an example), the sliding window size is 3 × 3, the orange area is the intersection area of the yellow and green areas, that is, the overlapping area of the two image patches.** The overlapping area, together with the yellow and green areas, respectively, constitutes the two image patches of $y_1^{pan}$ and $y_2^{pan}$.

(a)

| $P(\bar{\mathbf{Z}}\mid y_{v,(1,1)}^{pan})$ | $P(\bar{\mathbf{Z}}\mid y_{v,(1,2)}^{pan})$ | $P(\bar{\mathbf{Z}}\mid y_{v,(1,3)}^{pan})$ | | $P(\text{PAN}_{(1,1)})$ | $P(\text{PAN}_{(1,2)})$ | $P(\text{PAN}_{(1,3)})$ |
|---|---|---|---|---|---|---|
| $P(\bar{\mathbf{Z}}\mid y_{v,(2,1)}^{pan})$ | $P(\bar{\mathbf{Z}}\mid y_{v,(2,2)}^{pan})$ | $P(\bar{\mathbf{Z}}\mid y_{v,(2,3)}^{pan})$ | $=$ | $P(\text{PAN}_{(2,1)})$ | $P(\text{PAN}_{(2,2)})$ | $P(\text{PAN}_{(2,3)})$ |
| $P(\bar{\mathbf{Z}}\mid y_{v,(3,1)}^{pan})$ | $P(\bar{\mathbf{Z}}\mid y_{v,(3,2)}^{pan})$ | $P(\bar{\mathbf{Z}}\mid y_{v,(3,3)}^{pan})$ | | $P(\text{PAN}_{(3,1)})$ | $P(\text{PAN}_{(3,2)})$ | $P(\text{PAN}_{(3,3)})$ |

(a)

| $P(\bar{\mathbf{Z}}\mid y_{v,(1,1)}^{I})$ | $P(\bar{\mathbf{Z}}\mid y_{v,(1,2)}^{I})$ | $P(\bar{\mathbf{Z}}\mid y_{v,(1,3)}^{I})$ | | $P(\mathbf{I}_{(1,1)})^{o}$ | $P(\mathbf{I}_{(1,2)})^{o}$ | $P(\mathbf{I}_{(1,3)})^{o}$ |
|---|---|---|---|---|---|---|
| $P(\bar{\mathbf{Z}}\mid y_{v,(2,1)}^{I})$ | $P(\bar{\mathbf{Z}}\mid y_{v,(2,2)}^{I})$ | $P(\bar{\mathbf{Z}}\mid y_{v,(2,3)}^{I})$ | $=$ | $P(\mathbf{I}_{(2,1)})^{o}$ | $P(\mathbf{I}_{(2,2)})^{o}$ | $P(\mathbf{I}_{(2,3)})^{o}$ |
| $P(\bar{\mathbf{Z}}\mid y_{v,(3,1)}^{I})$ | $P(\bar{\mathbf{Z}}\mid y_{v,(3,2)}^{I})$ | $P(\bar{\mathbf{Z}}\mid y_{v,(3,3)}^{I})$ | | $P(\mathbf{I}_{(3,1)})^{o}$ | $P(\mathbf{I}_{(3,2)})^{o}$ | $P(\mathbf{I}_{(3,3)})^{o}$ |

(b)

**Fig 4. The corresponding computational process of probabilities** $P(\bar{\mathbf{Z}}\mid y_{v,(r,s)}^{pan})$ **and** $P(\bar{\mathbf{Z}}\mid y_{v,(r,s)}^{I})$ **in** $y_{v}^{pan}$ **and** $y_{v}^{I}$ **where** $v=1$.

$$P(\mathbf{I}_{(i,j)}\mid \bar{\mathbf{Z}}) = \frac{P(\bar{\mathbf{Z}}\mid \mathbf{I}_{(i,j)})P(\mathbf{I}_{(i,j)})}{\sum_{j=1}^{K}P(\bar{\mathbf{Z}}\mid I_{(i,j)})P(\mathbf{I}_{(i,j)})}$$

(5)

As a result, a new component, namely, $I^{new}$, can be described as follows:

$$I^{new}(i,j) = \begin{cases} \text{PAN}_{(i,j)}, & P(\text{PAN}_{(i,j)}\mid \bar{\mathbf{Z}}) \geq P(\mathbf{I}_{(i,j)}\mid \bar{\mathbf{Z}}) \\ I(i,j), & P(\text{PAN}_{(i,j)}\mid \bar{\mathbf{Z}}) < P(\mathbf{I}_{(i,j)}\mid \bar{\mathbf{Z}}) \end{cases}$$

(6)

Encouraging results are obtained on various datasets, which achieve better performance in terms of eight indices, including PSNR, RASE, RMSE, SAM, ERGAS $D_\lambda$, $D_S$, and QNR, than six state-of-the-art methods do. One issue in the proposed Bayesian decision rule in [28] is that not addressing the luminance relationship between different domain pixels affects its performance. To address this issue, in this work, we further analyze the ability of the Bayesian decision rule enhanced with cross-domain correspondence intensity modulation.

## 2.2 Cross-domain correspondence intensity modulation algorithm

Replacing $I$ component with $I^{new}$ can more effectively enhance the spatial resolution of MS image and reduce its spatial distortion than replacing $I$ component with the PAN image. However, the statistical attributes of Bayesian theory can only select pixels that the model considers optimal based on the magnitude of the posterior probability, and cannot handle inter-domain relationships between $I$ component with the PAN image. The obtained component $I^{new}$ is sub-local optimal solution. In this study, we consider the spatial differences between MS and PAN images to construct an intensity modulation coefficient to improve $I^{new}$. Specifically, we consider the intensity relationship between the pixels from the PAN and MS image domains and develop a cross-domain correspondence intensity modulation algorithm to modulate the pixel intensity in $I^{new}$ to produce the final intensity component $I^{improved}$ with improved pixel quality. The flowchart of the cross-domain correspondence intensity modulation algorithm is illustrated in Fig 5.

First, we use a Gaussian filter and a guided filter to extract the details of the I component of the MS image and the histogram-matched PAN image $P$ to obtain $M^{detail}$ and $PAN^{detail}$, respectively. Because $M^{detail}$ and $PAN^{detail}$ are usually used to represent the spatial contour texture information of the MS and PAN images, we use ($PAN^{detail} - M^{detail}$) to measure the scale of the change in intensity between the MS and PAN images, find the maximum pixel value in each

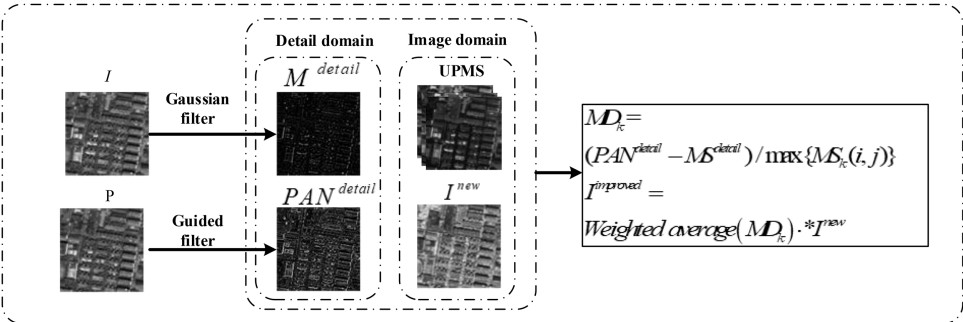

**Fig 5. Flowchart of cross-domain correspondence intensity modulation algorithm.**

MS band, and calculate $MD_k$ to assess the intensity relationship between the MS and PAN images. We believe that the components that satisfy this relationship are the ones desired by fusion. As a result, an improved $I^{new}$ component, namely, $I^{improved}$, can be described as follows:

$$I^{improved} = Weighted\ average(MD_k) \cdot I^{new} \tag{7}$$

where *Weighted average*$(\cdot)$ is the weighted average operation.

We perform an inverse transformation by replacing $I$ with $I^{improved}$ to obtain the pansharpened image as follows:

$$\overline{\mathbf{FMS}} = IHSto\overline{\mathbf{FMS}}(I^{improved}, H, S) \tag{8}$$

where $IHSto\overline{\mathbf{FMS}}(\cdot)$ is the inverse IHS transformation operation.

The resulting algorithm of cross-domain correspondence intensity modulation based on Bayesian decision-making for remote sensing image pansharpening is as follows: Algorithm 1.

```
Algorithm 1: Cross-domain correspondence intensity modulation based on bayesian-decision
Input: the low spatial resolution MS image LRMSₖ and the corresponding PAN image
Output: the pansharpened image FMSₖ
Begin
1) Upsample and interpolate LRMSₖ to obtain UPMSₖ
2) Histogram match PAN image to obtain P
3) Perform forward IHS transformation to obtain I component
          [I, H, S] = LRMStoIHS(LRMS)
4) Construct Bayesian probabilistic model to obtain Iⁿᵉʷ
          Iⁿᵉʷ(i, j) = { PA_N(i,j),   P(PA_N(i,j)|Z̄) ≥ P(I_(i,j)|Z̄)
                        { I(i, j),     P(PA_N(i,j)|Z̄) < P(I_(i,j)|Z̄)
5) Design cross-domain correspondence intensity modulation algorithm to obtain Iⁱᵐᵖʳᵒᵛᵉᵈ
          MDₖ = (PANᵈᵉᵗᵃⁱˡ − Mᵈᵉᵗᵃⁱˡ)/ max{MSₖ(i, j)}
          Iⁱᵐᵖʳᵒᵛᵉᵈ = Weighted average(MDₖ) · Iⁿᵉʷ
6) Perform an inverse IHS transformation to obtain the pansharpened image
          FMS = IHStoFMS(Iⁱᵐᵖʳᵒᵛᵉᵈ, H, S)
End
```

## 2.3 Performance testing of $I^{improved}$

In this study, we design a cross-domain correspondence intensity modulation algorithm to obtain a optimal intensity component $I^{improved}$ to attend the inverse IHS transformation instead of PAN image. The reason for doing this is that

$I^{improved}$ performs better than the PAN image. That is say the pansharpened results obtained by $IHStoFMS(I^{improved}, H, S)$ are better than that obtained by $IHStoFMS(PAN, H, S)$. The corresponding test results are shown in Figs 6 and 7, respectively. Fig 6(a)–6(c) are the ground truth, the pansharpened result by $IHStoFMS(PAN, H, S)$, and the pansharpened result by $IHStoFMS(I^{improved}, H, S)$, respectively. Normalize the quantification results of PSNR, UIQI, RASE, RMSE, SAM, and ERGAS. The used matrixes and the experimental data are from GeoEye datasets described in section III.A. From Fig 6, compared with Fig (a), Fig 6(c) has better spatial and spectral information than Fig 6(b). Furthermore, the Normalize the quantification results shown in Fig 7 show the performance of Fig 6(c) is better than Fig 6(b). In conclusion, the test results shown in Figs 6 and 7 are all to confirm the performance of $I^{improved}$ is better than that of the PAN image.

## 3. Experimental results and analysis

### 3.1 Experimental setup

In this section, the proposed method is evaluated with several datasets, including GeoEye1, WorldView-3, Pleiades, and WorldView-2. These datasets with three bands are sourced from http://www.kosmos-imagemall.com. The explicit information of the collected datasets is shown in Table 1. Five pairs of images are used for the fusion test, and a PAN image

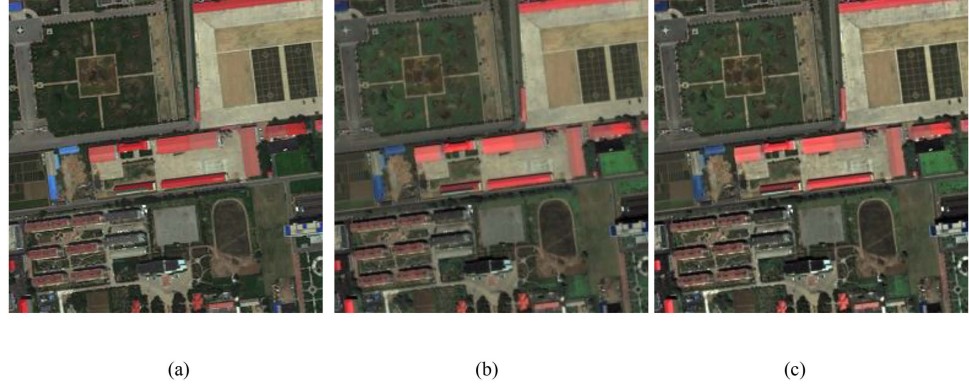

(a)                          (b)                          (c)

**Fig 6. Performance testing results.** (a) Ground truth, (b) the pansharpened result by $IHStoFMS(PAN, H, S)$, (c) the pansharpened result by $IHStoFMS(I^{improved}, H, S)$.

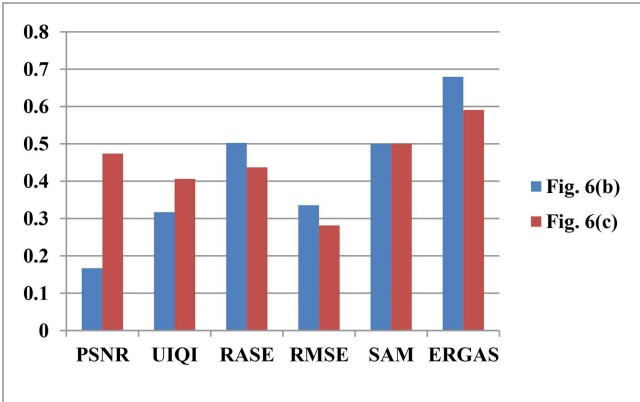

**Fig 7. Normalize the quantification results.**

**Table 1. Specifications of the datasets.**

| Test image | Spatial resolution (as meter) (PAN/MS) | Spectral resolutions | | | | PAN size (as pixel) | MS size (as pixel) | Ground truth (as pixel) |
|---|---|---|---|---|---|---|---|---|
| | | PAN | Spectral bands of MS (as nanometer) | | | | | |
| | | | Red | Blue | Green | | | |
| GeoEye | 0.46/1.64 | 450-800 | 655-690 | 450-510 | 510-580 | 256 × 256 | 64 × 64 × 3 | 256 × 256 × 3 |
| WorldView-3 | 0.31/1.24 | 397-454 | 626-696 | 445-517 | 507-586 | 256 × 256 | 64 × 64 × 3 | 256 × 256 × 3 |
| Pleiades | 0.5/2.0 | 470-830 | 590-710 | 500-620 | 430-550 | 256 × 256 | 64 × 64 × 3 | 256 × 256 × 3 |
| WorldView-2 | 0.5/2.0 | 450-800 | 630-690 | 510-580 | 450-510 | 256 × 256 | 64 × 64 × 3 | 256 × 256 × 3 |

and an MS image are included in each pair of data. Two types of comparison experiments are implemented in the following fusion exercises. One type is the simulated experiment with the reference image, in which the original images are degraded and decimated to 1/4. Following Wald's protocol [30], the reference image, called the ground truth, is the original MS image. The other type is a real experiment with no reference image, in which the PAN and MS images with original image attributes are fused and no image reference is used to evaluate the fused images. Eight widely used metrics [28] shown in Table 2, including RASE (↓), UIQI (↑), SAM (↓), ERGAS (↑), PSNR (↑), RMSE (↓), $D_\lambda$ (↓), $D_S$ (↓), and $QNR$ (↑), are employed for the following objective evaluation. Among them, the first 5 indicators are used in simulation experiments, and the last three indicators are used in real experiments. The symbol ↑ indicates that the higher the value is, the better the result, whereas ↓ means that the ideal value is 0.

Our work is traditional pansharpening method research. Various advanced and latest traditional pansharpening methods are compared with the proposed method. These methods include the adaptive IHS method [10], the band-dependent spatial detail (BDSD) injection-based method [31], BT [12], the robust BDSD (RBDSD)- based method [32], and the Bayesian decision-based fusion algorithm (BDFA) [28]. Furthermore, pansharpening by convolutional neural networks (PCNN) [33] is compared with the proposed method. All of the comparison methods used in this paper are open source codes offered by the corresponding authors.

## 3.2 Simulated and real experiments

**3.2.1 Simulated experiments.** In this study, we conduct simulation experiments on the GeoEye and WorldView-3 datasets. Figs 8–10 display the pansharpened results of three pairs of images, respectively. Figs 8(a)–10(a) are the reference images used as ground truths. Fig 8(b)–10(b) are the corresponding PAN images. Figs 8(c)-(i)–10(c)-(i) are the resulting images pansharpened by the different methods. To better evaluate the performance of the proposed method, in the qualitative evaluation of Figs 8–10, we choose an area to enlarge and compare the quality differences of the images pansharpened by different methods. The chosen area and the enlarged area are framed in red, and all the pansharpened images are compared with the reference image of the ground truth. The corresponding numerical evaluation results are

**Table 2. Assessment metrics with and without reference image.**

| Indices with reference image [25] | Indices without reference image [25] |
|---|---|
| Relative Average Spectral Error (RASE) | $D_\lambda$: the spatial distortion index |
| Spectral Angle Mapper (SAM) | |
| Universal Image Quality Indices (UIQI) | |
| Erreur Relative Global Adimensionnelle De Synthese (ERGAS) | |
| The peak signal-to-noise ratio (PSNR) | $D_S$: the spectral distortion index |
| Root Mean Square Error (RMSE) | QNR composed of $D_\lambda$ and $D_S$ |

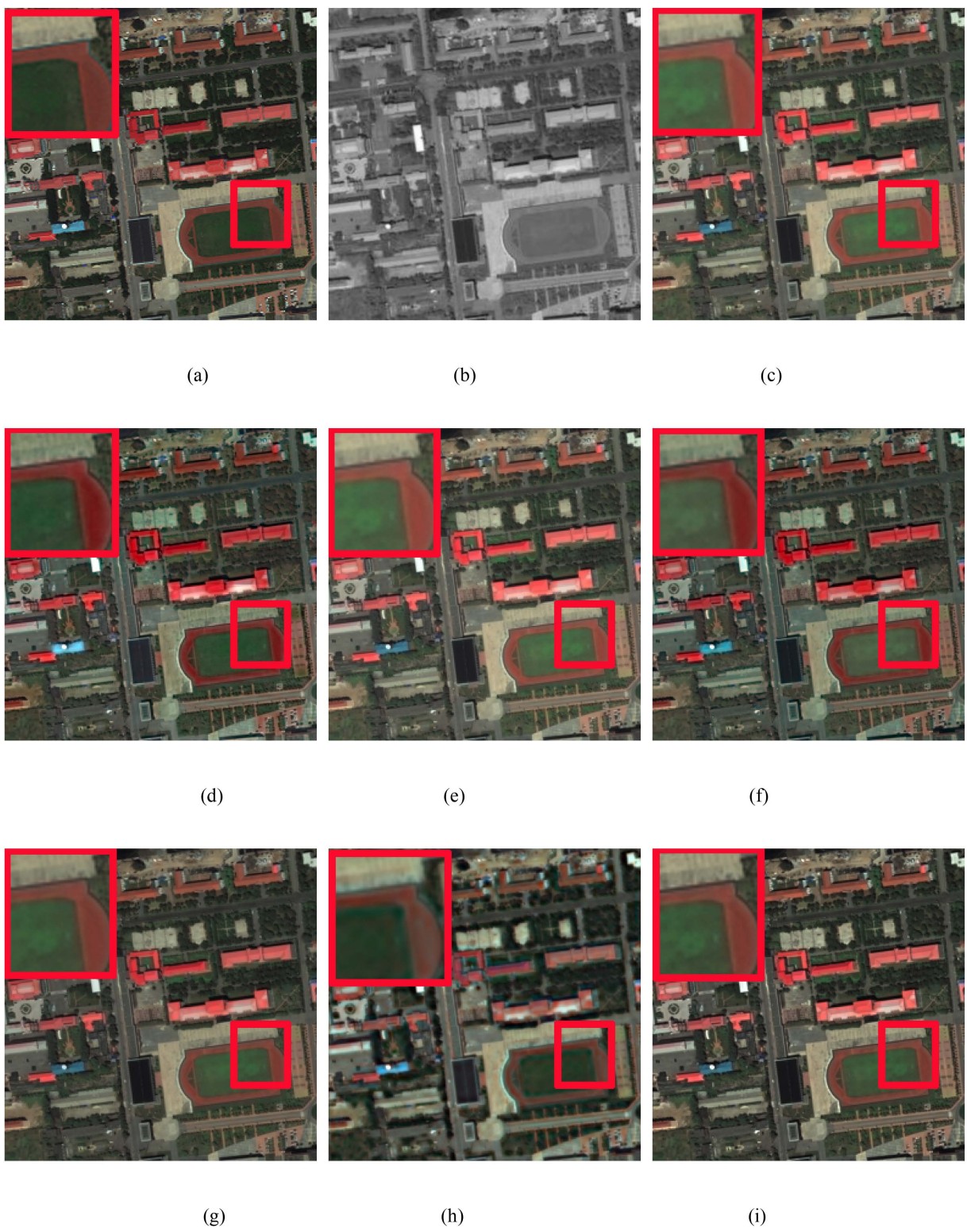

**Fig 8. GeoEye image fusion results.** (a) Ground truth, (b) PAN image, (c) IHS method, (d) BDSD method, (e) BT method, (f) RBDSD method, (g) BDBFA method, (h) PCNN method, (i) Proposed method.

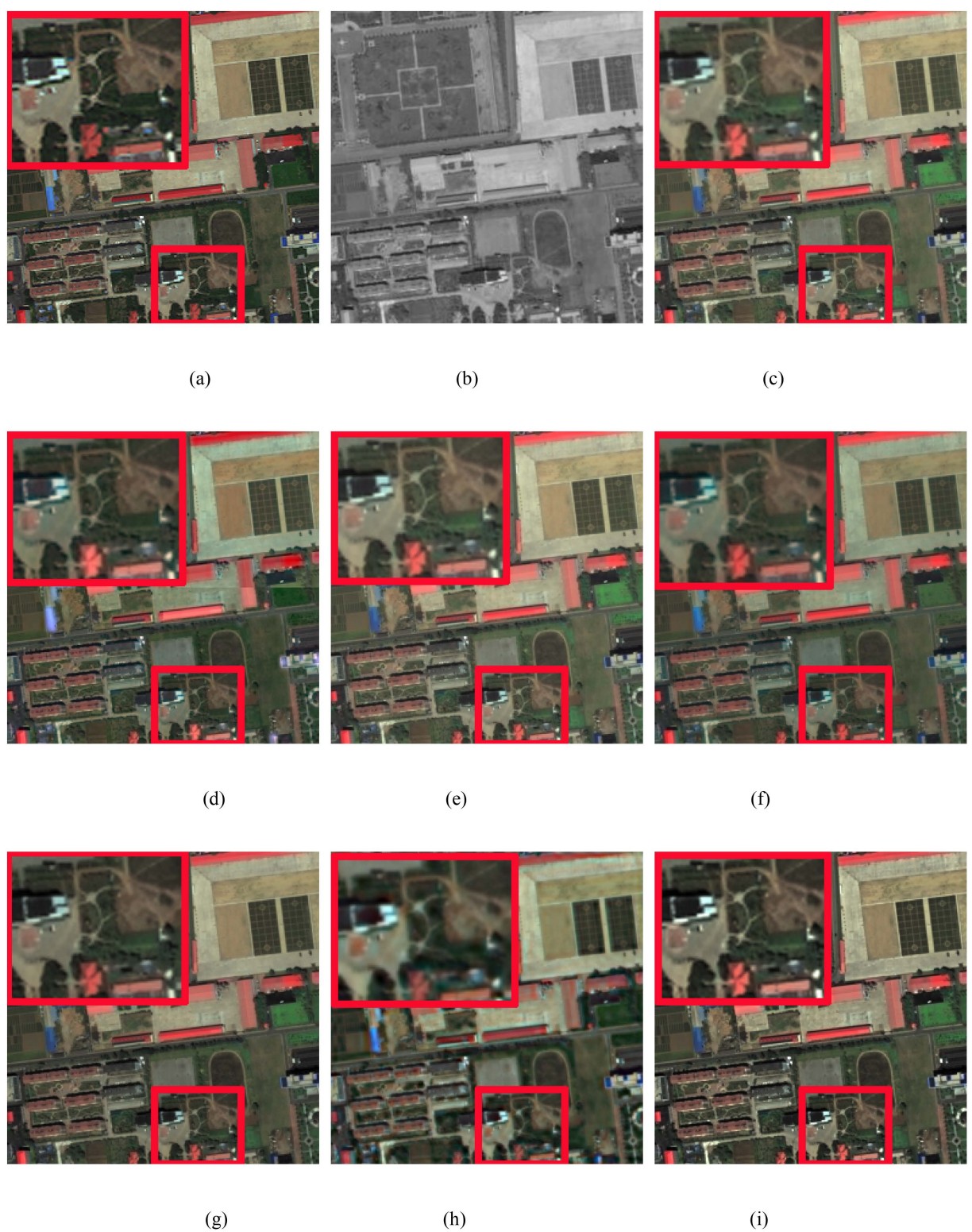

**Fig 9. GeoEye image fusion results.** (a) Ground truth, (b) PAN image, (c) IHS method, (d) BDSD method, (e) BT method, (f) RBDSD method, (g) BDBFA method, (h) PCNN method, (i) Proposed method.

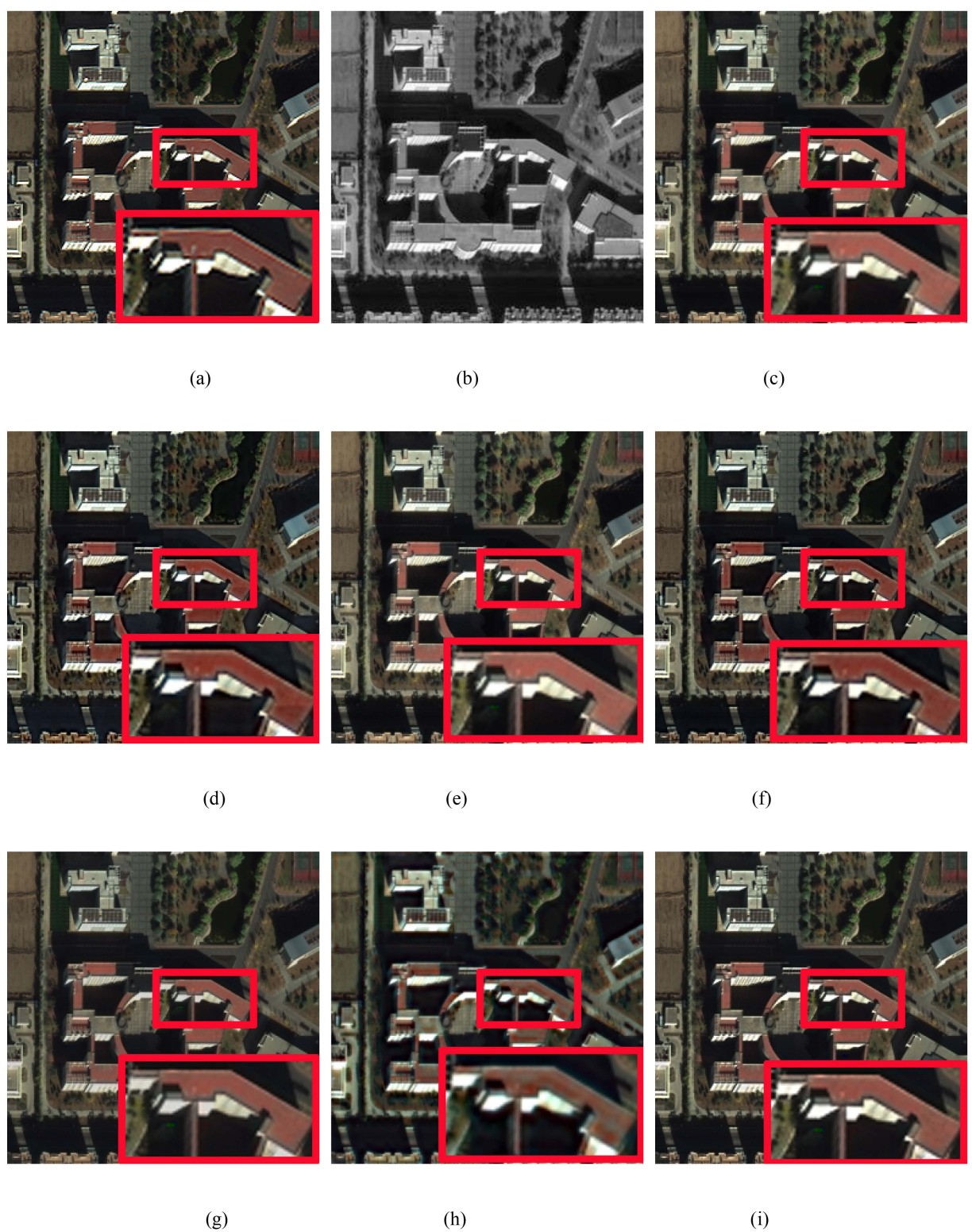

**Fig 10. WorldView-3 image fusion results.** (a) Ground truth, (b) PAN image, (c) IHS method, (d) BDSD method, (e) BT method, (f) RBDSD method, (g) BDBFA method, (h) PCNN method, (i) Proposed method.

shown in Table 3. In the quantitative assessment results of Table 3, the optimal value is shown in bold black, the second-best value is black italics in bold.

Fig 8 shows that the pansharpened image obtained via the IHS method has severe spectral distortion. The result produced by PCNN method has serious spectral distortions. The pansharpened images produced by the BDSD and RBDSD methods have obvious spectral distortions, which are shown in white and red. The resulting image pansharpened by the BT method is better than that of the IHS method, but compared with those of the BDSD and RBDSD methods, the result of the BT method has more severe spectral distortion. The pansharpened images produced by BDFA and the proposed method are close to the ground truth in terms of spectral characteristics. However, the spatial quality of the resulting image obtained by the proposed method is better than that the BDFA. In addition, from Table 3, comapried with the latest advanced traditional pansharpening methods, the numerical evaluation results obtained by the proposed method are better than those of the other methods in terms of the PSNR, UIQI, RASE, RMSE, and ERGAS. Although the corresponding value obtained with the SAM is not the best, it is close to the best value. Add the comparision of the latest deep learning-based pansharpening method PCNN, we can found that our method has the best values in PSNR, SAM, and ERGAS, the second best values in UIQI, RASE, and RMSE, and the ranking of SAM value remains unchanged.

As shown in Fig 9, the enlarged red rectangles show that the pansharpened images produced by the IHS, BT, and RBDSD

methods are blurry and have obvious spectral distortion. The result produced by PCNN method has obvious spectral and spatial distortions. The results of the BDSD and BDFA methods have better visual effects than those of the IHS, BT, and RBDSD methods, but the BDSD results have obvious spectral distortion, as shown in white, and the

**Table 3. Quantitative evaluation results.**

| Data | Method | Indices with reference image | | | | | |
|------|--------|------|------|------|------|------|------|
| | | PSNR | UIQI | RASE | RMSE | SAM | ERGAS |
| Fig 8 (GeoEye) | IHS | 21.8171 | 0.9223 | 27.6131 | 20.6865 | **3.4558** | 5.7559 |
| | BDSD | 23.7959 | 0.9144 | 21.9874 | 16.4720 | 5.2310 | 5.4254 |
| | BT | 21.8163 | 0.9223 | 27.6155 | 20.6883 | *3.4560* | 5.7562 |
| | RBDSD | 23.4441 | 0.9052 | 22.8962 | 17.1528 | 5.1435 | 5.6045 |
| | BDFA | *24.6575* | 0.9245 | 19.9110 | 14.9164 | 3.4590 | 5.0190 |
| | PCNN | 22.9959 | **0.9477** | **18.8462** | **14.1187** | 5.5058 | *4.8430* |
| | Proposed | **24.8496** | *0.9374* | *19.4755* | *14.5902* | 3.4578 | **4.8225** |
| Fig 9 (GeoEye) | IHS | 20.8579 | 0.9187 | 27.8717 | 23.1016 | *2.9967* | 5.7577 |
| | BDSD | 23.5343 | 0.9279 | 20.4807 | 16.9756 | 4.1920 | 5.0408 |
| | BT | 20.8575 | 0.9187 | 27.8730 | 23.1028 | **2.9966** | 5.7579 |
| | RBDSD | 23.8543 | 0.9178 | 19.7399 | 16.3616 | 3.6574 | 4.7660 |
| | BDFA | *24.4169* | 0.9317 | 18.5019 | 15.3354 | 2.9987 | 4.6792 |
| | PCNN | 22.8216 | **0.9540** | **16.6955** | **13.8382** | 4.4408 | **4.3313** |
| | Proposed | **24.4474** | *0.9406* | *18.4370* | *15.2816* | 3.0005 | *4.5909* |
| Fig 10 (WorldView-3) | IHS | 22.8270 | 0.9474 | 30.4173 | 18.4157 | 3.8972 | *6.9800* |
| | BDSD | 23.5226 | *0.9521* | *28.0764* | *16.9984* | 4.8331 | 7.1592 |
| | BT | 22.8266 | 0.9474 | 30.4187 | 18.4166 | 3.8970 | 6.9801 |
| | RBDSD | 23.4412 | **0.9535** | 28.3408 | 17.1586 | 4.6478 | 7.2035 |
| | BDFA | 23.1096 | 0.9346 | 29.4438 | 17.8263 | *3.8932* | 7.4521 |
| | PCNN | **24.9255** | 0.9481 | 29.1212 | 17.6310 | 6.4041 | 7.4778 |
| | Proposed | *23.6300* | 0.9496 | **27.7314** | **16.7896** | **3.8891** | **6.8412** |

spatial resolutions of the resulting images obtained by the BDSD and BDFA methods are significantly lower than those of the proposed method. The proposed method produces the best pansharpened image both spatially and spectrally compared with all the other comparison methods. In addition, as shown in Table 3, similar to the results in Fig 8, comapried with the latest advanced traditional pansharpening methods, the numerical evaluation results obtained by the proposed method are better than the results of the comparison methods in terms of the PSNR, UIQI, RASE, RMSE, and ERGAS. Although the corresponding value obtained via the SAM is not the best, it is close to the best value. Add the comparision of the latest deep learning-based pansharpening method PCNN, we can found that our method has the best values in PSNR, the second best values in UIQI, RASE, RMSE, and ERGAS, and the ranking of SAM value remains unchanged.

Fig 10 shows that the pansharpened images obtained via the IHS and BT methods have severe spectral distortion. The result produced by PCNN method has obvious spectral and spatial distortions. As seen from the enlarged red rectangle, thepansharpened results produced by BDSD and RBDSD have obvious spectral distortions, as shown in red. Similar to Fig 8, the pansharpened images produced by the BDFA and the proposed method are close to the ground truth spectral data. However, the spatial quality of the resulting image obtained by the proposed method is significantly higher than that of the BDFA. In addition, as shown in Table 3 (Figs 8–10), comapried with the latest advanced traditional pansharpening methods, the numerical evaluation results obtained by the proposed method are better than those of the other methods in terms of the PSNR, RASE, RMSE, SAM, and ERGAS. Although the corresponding UIQI value is not the best, it is close to the best value. Add the comparison of the latest deep learning-based pansharpening method PCNN, we can found that our method has the best values in RASE, RMSE, SAM, and ERGAS, the second best values in PSNR, and the ranking of UIQI value remains unchanged.

Comprehensively evaluate the experimental results in both qualitative and quantitative aspects, the performance of the proposed method is significantly better than that of the comparison methods in all the simulated experiments. Meanwhile, the following point is worth noting. Since the information with a lower spatial resolution compared to that of the PAN image or redundant information in $I$ component are discarded in the Bayesian decision-making stage. The inter-domain relationships caused by these discarded information cannot be repaired by the proposed cross-domain correspondence intensity modulation algorithm. The spectral distortion of neighboring pixels caused by the intensity change of a single pixel is inevitable. SAM measures the degree of spectral information proximity between two images in local. The SAM metric of the proposed method not optimal show our method's ability to handle local spectral distortion is not optimal. In the future, we will further study and strive to solve this problem.

**3.2.2 Real experiments.** In this study, we conduct real experiments on the Pleiades and WorldView-2 datasets. Figs 11 and 12 display the pansharpened results of two pairs of images. Figs 11(a) and 12(a) show the MS images. Figs 11(b) and 12(b) show the corresponding PAN images. Figs 11(c)-(i) and 12 (c)-(i) show the resulting images pansharpened by the different methods. To better evaluate the performance of the proposed method, in the qualitative evaluation of Figs 11 and 12, we choose an area to enlarge and compare the quality differences of the images pansharpened by different methods. The corresponding numerical evaluation results are shown in Table 4. In the quantitative assessment results of Table 3, the optimal value is shown in bold black, the second-best value is black italics in bold.

As shown in Fig 11, the result produced by PCNN method has obvious spectral and spatial distortions. The enlarged red rectangles show that the pansharpened images produced by the proposed method achieve the best spectral and spatial quality, especially in terms of white and blue protection, compared with the IHS, BT, BDSD, RBDSD, and BDFA methods. In addition, it is well known that the QNR is composed of $D_\lambda$ and $D_S$, and the higher the value is, the better the result. As shown in Table 4, comapried with the latest advanced traditional pansharpening methods, although the proposed method does not yield the best $D_\lambda$ and $D_S$ values, it achieves the best QNR value. Add the comparison of the latest deep learning-based pansharpening method PCNN, we can found that our method has the best values in $D_S$, the second best values in QNR.

 

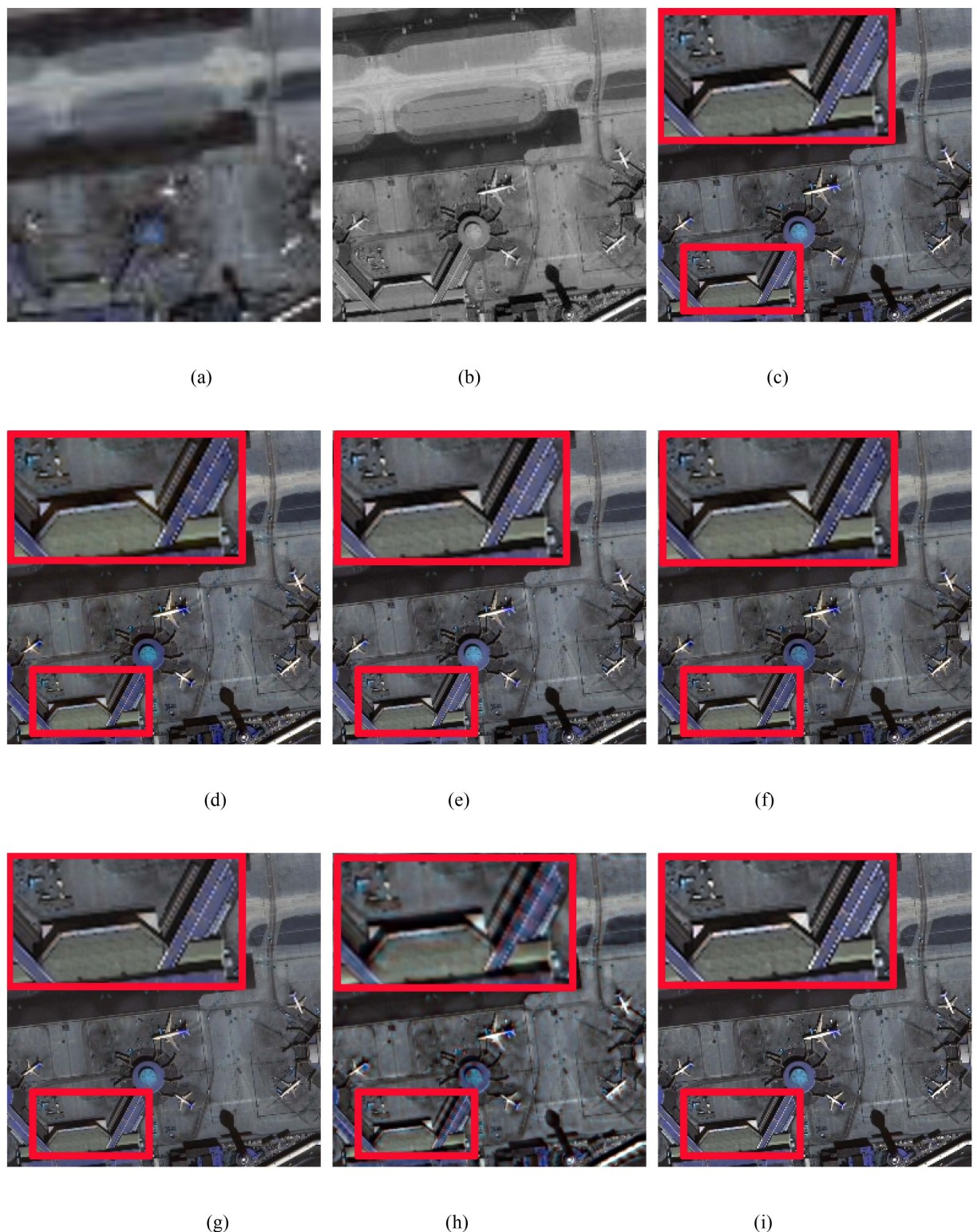

(a) (b) (c)

(d) (e) (f)

(g) (h) (i)

**Fig 11. Pleiades image fusion results.** (a) LRMS image, (b) PAN image, (c) IHS method, (d) BDSD method, (e) BT method, (f) RBDSD method, (g) BDBFA method, (h) PCNN method, (i) Proposed method.

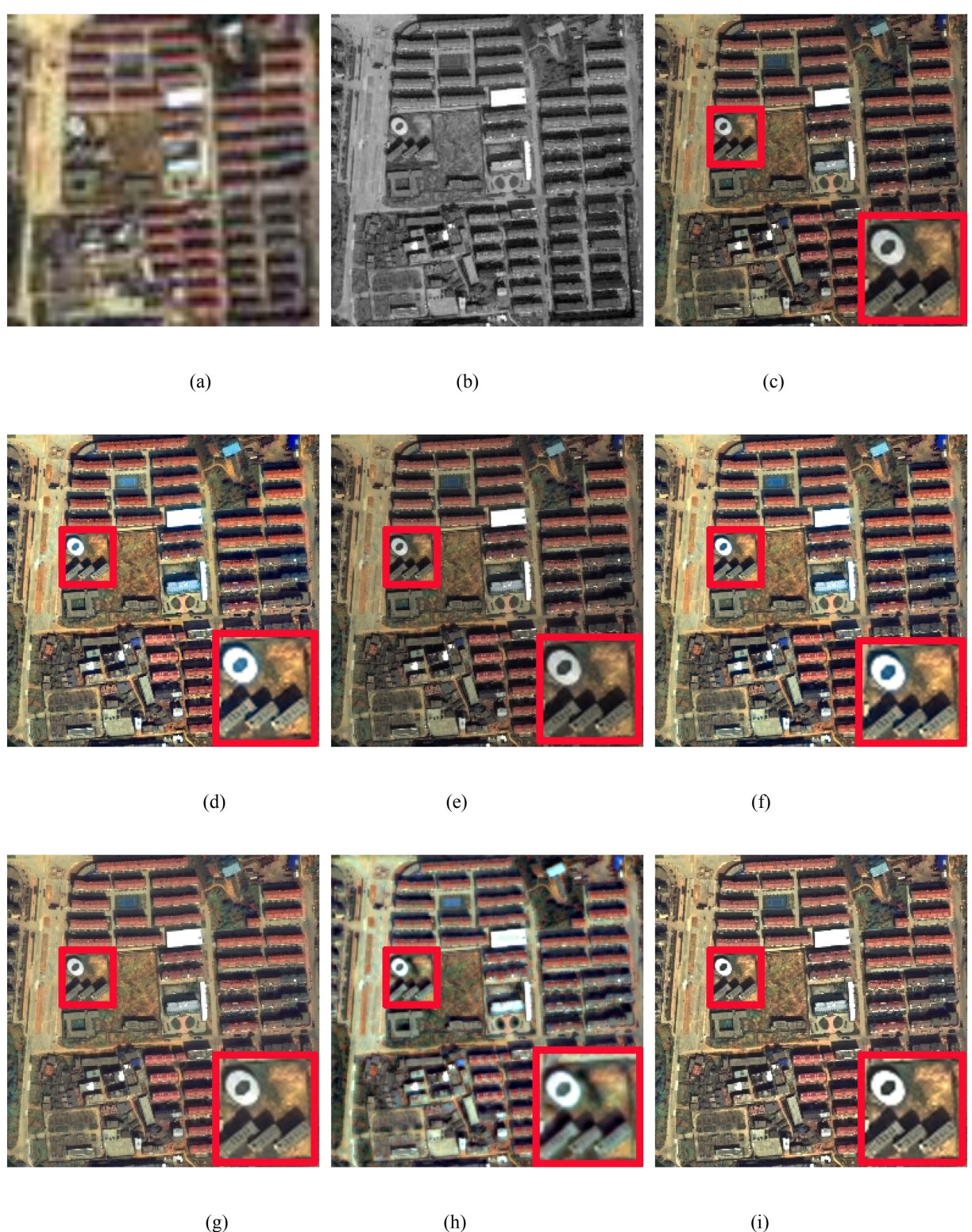

**Fig 12. WorldView-2 image fusion results.** (a) LRMS image, (b) PAN image, (c) IHS method, (d) BDSD method, (e) BT method, (f) RBDSD method, (g) BDBFA method, (h) PCNN method, (i) Proposed method.

**Table 4. Quantitative evaluation results.**

| Methods | Fig 11 (Pleiades) | | | Fig 12 (WorldView-2) | | |
|---|---|---|---|---|---|---|
| | $D_\lambda$ | $D_S$ | QNR | $D_\lambda$ | $D_S$ | QNR |
| IHS | 0.0045 | 0.0793 | 0.9159 | 0.0240 | 0.1530 | 0.8270 |
| BDSD | 0.0085 | 0.0684 | 0.9237 | 0.0356 | *0.0704* | 0.8965 |
| BT | *0.0048* | 0.0791 | 0.9164 | 0.0239 | 0.1529 | 0.8269 |
| RBDSD | **0.0022** | 0.0743 | 0.9237 | 0.0331 | 0.0830 | 0.8866 |
| BDFA | 0.0092 | *0.0661* | 0.9253 | *0.0111* | 0.1095 | 0.8806 |
| PNN | 0.0163 | **0.0356** | **0.9487** | 0.0341 | **0.0525** | **0.9153** |
| Proposed | 0.0076 | 0.0673 | *0.9256* | **0.0110** | 0.0909 | *0.8991* |

As shown in Fig 12, the result produced by PCNN method has obvious spectral and spatial distortions. The IHS and BT methods produce pansharpened images with severe global spectral distortion. As the elliptic pattern region in the enlarged red rectangles shows, the BDSD and RBDSD methods produce pansharpened images with severe local spectral distortion. BDFA and the proposed method achieve better image results than the other methods do. In contrast, between the pansharpened images obtained by BDFA and the proposed method, the global spectra and spatial qualities of the two resulting images are approximate; however, the resulting image obtained by the proposed method is clearly better in the enlarged red rectangles than that of BDFA. Moreover, as shown in Table 4, comapried with the latest advanced traditional pansharpening methods, our method achieves the best $D_\lambda$ and QNR values and the second best $D_S$ value. Add the comparision of the latest deep learning-based pansharpening method PCNN, we can found that our method has the best values in $D_\lambda$, the second best values in QNR.

To sum up, the proposed method considers the spatial differences between the images before fusion. The fluctuation amplitude of the intensity before and after image fusion is measured by the ratio of the spatial detail difference between the different domains of PAN and MS images to the maximum pixel value in MS image. In the proposed modulation algorithm, this rate is applied as the core technology to modulate the intensity component in the fused image, making it approximate the intensity component in the expected image, which effectively reduces the spatial distortion of the fused image. Furthermore, in the IHS inverse transformation, original H and S component are kept unchanged while the improved I component is applied, which effectively reduces the spectral distortion of the fused image. Comprehensively evaluate the experimental results in both qualitative and quantitative aspects, the performance of the proposed method is significantly better than that of the comparison methods in all real experiments.

### 3.3 Computational efficiency analysis

In this section, we handle the computational efficiency analysis. We calculate the average consuming time of five groups fusion images obtained in this paper for all the methods. The results were computed and listed in Table 5.

From Table 5, we can find the network of PCNN requires a long time to train that results in the running time and resource consumption are great. IHS and BT belong to the early CS based methods. The characteristics of them are simplicity, fast, but serious spectral distortion. BDSD takes into account the image context information that results in the algorithm is relatively complex. RBDSD is the improved version of the BDSD method. So, the run time is longer than that of the the BDSD method. BDFA and the proposed method constructed bayesian probabilistic model to calculate the probability that each pixel in the image is selected, which has increased the complexity of the algorithm. Compared with PCNN, the computational efficiency is high. However, the running time is longer than other latest advanced traditional pansharpening methods, which further improvements are needed in the future.

**Table 5. The average consuming time of the comparison methods.**

| Methods | Time (second) | Remark |
|---|---|---|
| IHS | 0.0871 | The early CS method had a simple algorithm |
| BDSD | 0.31 | It takes into account the context information in an image, which results in its algorithm is relatively complex. |
| BT | 0.0652 | The early CS method had a simple algorithm. |
| RBDSD | 0.444 | The improved version of the BDSD method. |
| BDFA | 2.6614 | The use of Bayesian probability models increases the complexity of the algorithm. |
| PCNN | 0.4369 | The network training time exceeds 24 hours |
| Proposed | 2.7957 | The use of Bayesian probability models increases the complexity of the algorithm |

## 4. Conclusion

In this paper, a novel approach in which cross domain correspondence intensity modulation is based on a bayesian-decision for remote sensing image parsharpening is proposed. In the proposed method, an IHS transformation is used to extract the I component in the MS image. A Bayesian probabilistic model is proposed for the fusion of the I component and the corresponding PAN image. An observation model based on Bayesian estimation is first established for remote sensing image fusion. Selection of only the correct pixels on the basis of the Bayesian probabilistic model is missing for the modulation of the relationship between pixels in different domains, which easily causes spectral and spatial distortion of the pansharpened image. In the context of this question, a cross-domain correspondence intensity modulation algorithm is proposed to improve the fusion result of the I component and the corresponding PAN image based on the Bayesian probabilistic model to provide a solution. Finally, an IHS inverse transformation is implemented to obtain the pansharpened image. The experimental results on degraded and real images from the GeoEye, WorldView-3, Pleiades, and WorldView-2 datasets confirm that the performance of the proposed method exceeds that of several related and popular existing fusion methods.

## Supporting information

**S1 Data. Experimental data used in this paper.**
(RAR)

## Author contributions

**Data curation:** Xunyan Jiang, Jinhua Liu.

**Formal analysis:** Zhaosheng Xu.

**Investigation:** Xunyan Jiang.

**Methodology:** Lei Wu.

**Project administration:** Lei Wu.

**Resources:** Lei Wu, Xunyan Jiang.

**Software:** Lei Wu, Zhijian Zhao.

**Supervision:** Xunyan Jiang.

**Writing – original draft:** Lei Wu.

**Writing – review & editing:** Zhijian Zhao.

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
