## [Decision Letter · Decision Letter 0]

27 Jun 2025

Dear Dr. Wu,

State of the art: Expand the introduction and related work sections to include recent pansharpening methods, particularly deep learning approaches, ensure that you incorporate more diverse recent research (avoiding self citations) and include specific methods suggested by the reviewers.Experiments: Add comprehensive comparisons with latest deep learning-based pansharpening methods (CNN, GAN, Transformer-based models), and include performance analysis beyond just results description, explaining why the proposed method achieves superior performance. Perform an analysis to address the suboptimal SAM metric performance. Add computational efficiency analysis including running time and resource consumption.Presentation and language: Perform thorough proofreading to address grammatical errors and awkward sentence constructions throughout. Address specific issues, e.g., "twofold" should be "threefold" and "Bayesian-making" phrasing. Ensure consistent symbol formatting (e.g., italic vs non-italic forms of symbol I)Methodology: Provide clearer explanation of the Bayesian probabilistic model computational process, especially sliding window technique details. Clarify the relationship between image patch (V) and image size. Better describe the z_hat concept and its application in formulas 5-6. Explain probability computation methods in equations 5-6 and address the position vs patch probability inconsistency. Articulate the theoretical advancements beyond the previous BDFA work to establish novelty.

We look forward to receiving your revised manuscript.

Kind regards,

Panos Liatsis, PhD

Academic Editor

PLOS ONE

Journal Requirements:

3. Thank you for stating the following financial disclosure: [This work is supported by National Natural Science Foundation of China (No. 62441112 and No. 72461031), by Jiangxi Provincial Natural Science Foundation (No. 20232BAB201026), and by Science and Technology Re-search Project of Jiangxi Provincial Department of Education (No. GJJ2402102).]. 

Please state what role the funders took in the study.  If the funders had no role, please state: " "The funders had no role in study design, data collection and analysis, decision to publish, or preparation of the manuscript." "

Reviewers' comments:

Reviewer's Responses to Questions

**Comments to the Author**

1. Is the manuscript technically sound, and do the data support the conclusions?

Reviewer #1: Yes

Reviewer #2: Yes

Reviewer #3: Yes

Reviewer #4: Yes

2. Has the statistical analysis been performed appropriately and rigorously?

Reviewer #1: Yes

Reviewer #2: Yes

Reviewer #3: Yes

Reviewer #4: Yes

3. Have the authors made all data underlying the findings in their manuscript fully available?

Reviewer #1: Yes

Reviewer #2: Yes

Reviewer #3: No

Reviewer #4: Yes

4. Is the manuscript presented in an intelligible fashion and written in standard English?

Reviewer #1: Yes

Reviewer #2: Yes

Reviewer #3: Yes

Reviewer #4: Yes

Reviewer #1: This manuscript proposes a novel remote sensing image fusion algorithm and validates its effectiveness on GeoEye1, WorldView-3, Pleiades, and WorldView-2 datasets. Overall, this research is innovative and promising for application, but there are still some problems that need to be solved and improved by the authors.

1. In Fig1, the "IHS" in front of "I" should be changed to "IHS forward Transform".

2. In Section "B. Fusion Algorithm Based on Bayesian Decision", it is recommended to explain the relationship between the image patch (V) and the image size for better clarity.

3.The meanings of z_hat in Fig. 1, referred to as the " the class of the fused I component". In Formula 5 and 6, z_hat is applying. It's advisable to descript the concept in this part.

4. Maintain consistency in symbol formatting throughout the entire text. For example, in Section "A. Bayesian probabilistic model for image component fusion", the symbol I is presented in italic and non-italic forms.

5. In (5) and (6), the left side of equation is probability in position (i,j) while in the right side, there are probabilities in the patches at whole positions. Why?

6. How are probabilities in (5)-(6) computed?

7. In the IV Experimental Results and Analysis section of the manuscript, only a description of the results is given, with no analysis or further explanation of the reasons why the proposed method achieves superior performance.

8. The introduction and related work sections do not sufficiently cover the state-of-the-art (SOTA) methods. The authors are encouraged to include and discuss additional pan-sharpening methods, as outlined below:

MSAN: Multiscale self-attention network for pansharpening, DOI: 10.1016/j.patcog.2025.111441.

A Unified Pansharpening Model Based on Band-Adaptive Gradient and Detail Correction, DOI:10.1109/TIP.2021.3137020.

AWFLN: An Adaptive Weighted Feature Learning Network for Pansharpening, DOI: 10.1109/TGRS.2023.3241643.

Intensity mixture and band-adaptive detail fusion for pansharpening, DOI: https://doi.org/10.1016/j.patcog.2023.109434.

Invertible Attention-Guided Adaptive Convolution and Dual-Domain Transformer for Pansharpening, DOI: 10.1109/JSTARS.2025.3531353

Cross-Scale Interaction With Spatial-Spectral Enhanced Window Attention for Pansharpening, DOI: 10.1109/JSTARS.2024.3413856.

Reviewer #2: 1. It is recommended that the computational process in the Bayesian probabilistic model be further elaborated, in particular the details of the treatment of overlapping regions in the sliding window technique.

2. The paper compares the method with some traditional deep learning methods and Bayesian modeling methods, but lacks a detailed comparison with the latest you deep learning methods, and suggests adding comparison experiments with the latest deep learning methods.

3. The running efficiency and resource consumption of the algorithm are not mentioned in the paper, and it is suggested that the authors add relevant experimental analyses to show the advantages and disadvantages of the method in terms of efficiency.

4. It is suggested that the flowchart of Figure 2 be further optimized, such as adding textual annotations for key steps, to make it easier for readers to understand.

Reviewer #3: According to the description, this manuscript proposed a method for remote sensing image pansharpening based on Bayesian making with wross-domain correspondence intensity modulation. Essentially, this is still an improved IHS method, but yielding a new computing framework. Extensive experiments proved the effectiveness of the proposed method. Please see the following comments.

[Q1] What are the advantages of the new intensity component I^{improved} generated from PAN image, compared with PAN image?

[Q2] Some writing issues need to be noted, e.g., “twofold” should be “threefold”. Please check others carefully!

[Q3] The self-citation rate of the manuscript is quite significant. It is recommended that the latest research in the field of pansharpening be thoroughly investigated. For instance, for the model-based methods, the following works are related:

[R1] LRTCFPan: Low-Rank Tensor Completion Based Framework for Pansharpening, IEEE TIP, 2023.

[R2] Pansharpening via Semi-Framelet-Guided Sparse Reconstruction, IF, 2024.

[R3] A Novel Spatial Fidelity with Learnable Nonlinear Mapping for Panchromatic Sharpening, IEEE TGRS, 2023.

[Q4] Similar to [Q3], the recently proposed methods are recommended for comparison in experiments, to demonstrate the validity of the proposed method.

[Q5] Why is the SAM metric of the proposed method not optimal?

Reviewer #4: Comments

Strengths:

1. Novelty and Technical Contribution:

o The paper presents a novel Bayesian-based framework incorporating cross-domain correspondence intensity modulation for remote sensing image pansharpening. The use of both probabilistic modeling and domain-specific modulation provides a unique fusion strategy that appears to be well-motivated and methodologically sound.

2. Experimental Rigor:

o The paper is thorough in its experimental setup. It uses both simulated and real datasets (GeoEye, WorldView-2/3, Pleiades) and evaluates the method using a comprehensive set of metrics (PSNR, UIQI, SAM, ERGAS, RASE, RMSE, Dλ, SD, QNR), which is commendable.

3. Performance Superiority:

o The proposed method consistently outperforms several state-of-the-art methods, including IHS, BDSD, BT, RBDSD, and the authors’ prior work (BDFA), in both qualitative (visual inspection) and quantitative evaluations.

4. Clear Visuals and Analysis:

o Figures 4–8 clearly illustrate the effectiveness of the method, with well-chosen zoomed-in areas that emphasize improvements in both spatial and spectral fidelity.

Weaknesses:

1. Clarity and Language:

o While the technical content is solid, the manuscript suffers from poor grammar and awkward sentence constructions throughout. For example, “cross-domain correspondence intensity modulation based on Bayesian-making” is grammatically incorrect and should be rephrased for clarity. Many such phrasings appear throughout the manuscript and impede readability.

2. Over-Emphasis on Prior Work:

o The method builds heavily on the previously published Bayesian Decision Fusion Algorithm (BDFA), with the novelty primarily residing in the post-processing modulation step. This might limit the paper’s overall novelty unless the authors better articulate the theoretical advancements and differences.

3. Lack of Deep Learning Baseline Comparisons:

o The paper does not include comparisons with recent deep learning-based pansharpening techniques, which are currently the state-of-the-art in many remote sensing tasks. Including such baselines (e.g., CNN, GAN, Transformer-based models) would significantly strengthen the experimental section and relevance.

4. Insufficient Theoretical Justification:

o Although the Bayesian probabilistic model and modulation algorithm are described in detail, a more rigorous theoretical analysis of convergence, complexity, or statistical consistency would add depth to the methodology.

5. Figure Captions and Tables:

o Some figures and tables could benefit from clearer labeling and more informative captions. For example, the abbreviations (e.g., RASE, Dλ) are not always explained immediately in context, requiring readers to refer back to earlier sections or tables.

**Do you want your identity to be public for this peer review?** For information about this choice, including consent withdrawal, please see our Privacy Policy

Reviewer #1: No

Reviewer #2: No

Reviewer #3: No

Reviewer #4: **Yes: ** Subbiah Manthira Moorthi

---

## [Author Response · Author response to Decision Letter 1]

14 Aug 2025

We have responded to the comments from the reviewers and the editors one by one. Detailed information has been uploaded to the "Attach Files" in the form of an attachment. Please click on the decision letter link for the specific reviewers and editors. Please find the "Respond to reviewers" attachment and view it.

---

## [Decision Letter · Decision Letter 1]

5 Sep 2025

Dear Dr. Wu,

Thank you for submitting your manuscript to PLOS ONE. After careful consideration, we feel that it has merit but does not fully meet PLOS ONE’s publication criteria as it currently stands. Therefore, we invite you to submit a revised version of the manuscript that addresses the points raised during the review process.

We look forward to receiving your revised manuscript.

Kind regards,

Yaseen Al-Mulla

Academic Editor

PLOS ONE

Journal Requirements:

Reviewers' comments:

Reviewer's Responses to Questions

**Comments to the Author**

Reviewer #1: All comments have been addressed

Reviewer #2: All comments have been addressed

2. Is the manuscript technically sound, and do the data support the conclusions?

Reviewer #1: Yes

Reviewer #2: Yes

3. Has the statistical analysis been performed appropriately and rigorously?

Reviewer #1: Yes

Reviewer #2: Yes

4. Have the authors made all data underlying the findings in their manuscript fully available?

Reviewer #1: Yes

Reviewer #2: Yes

5. Is the manuscript presented in an intelligible fashion and written in standard English?

Reviewer #1: Yes

Reviewer #2: Yes

Reviewer #1: The manuscript proposes a cross-domain correspondence intensity modulation framework guided by a Bayesian-decision mechanism for remote sensing pan sharpening. The goal is to enhance spatial detail transfer from PAN while preserving spectral fidelity in MS, especially under domain shifts across sensors or scenes. The authors have satisfactorily addressed all comments; acceptance is recommended.

Reviewer #2: Thank you for the author's response to my previous questions. Most of the issues have been addressed, but I still have a few minor suggestions:

1. For Fig. 8, 9, and 10, it is recommended to include the original multispectral and panchromatic images in addition to the ground truth. This would allow readers to independently assess whether certain textural information has been lost, as the advantage of your method is not particularly evident in the provided examples. The same suggestion applies to other figures—supplementing them with original imagery would be beneficial.

2.In Table 5, the "Remark" column only includes comments on the PCNN. It would be helpful if remarks regarding other methods could also be added for clarity and comparison.

3.The Conclusion section currently contains content that is more appropriate for the Discussion section. It is recommended to separate these two parts to ensure a clearer structure.

**Do you want your identity to be public for this peer review?** For information about this choice, including consent withdrawal, please see our Privacy Policy

Reviewer #1: No

Reviewer #2: No

---

## [Author Response · Author response to Decision Letter 2]

8 Oct 2025

We have responded to specific reviewer and editor comments by uploading the attached files of “Response to Reviewers”.

---

## [Editor Report · Decision Letter 2]

12 Oct 2025

Cross-domain correspondence intensity modulation based on bayesian-decision for remote sensing image pansharpening

PONE-D-25-22687R2

Dear Dr. Wu,

We’re pleased to inform you that your manuscript has been judged scientifically suitable for publication and will be formally accepted for publication once it meets all outstanding technical requirements.

Kind regards,

Yaseen Al-Mulla

Academic Editor

PLOS ONE
---

## [Editor Report · Acceptance letter]

PONE-D-25-22687R2

PLOS ONE

Dear Dr. Wu,

I'm pleased to inform you that your manuscript has been deemed suitable for publication in PLOS ONE. Congratulations! Your manuscript is now being handed over to our production team.

Kind regards,

on behalf of

Dr. Yaseen Ahmed Al-Mulla

Academic Editor

PLOS ONE